# Incidence of Acute Upper Gastrointestinal Bleeding and Related Risk Factors among Elderly Patients Undergoing Surgery for Major Limb Fractures: An Analytical Cohort Study

**DOI:** 10.3390/healthcare11212853

**Published:** 2023-10-30

**Authors:** Guan-Yu Chen, Wen-Tien Wu, Ru-Ping Lee, Ing-Ho Chen, Tzai-Chiu Yu, Jen-Hung Wang, Kuang-Ting Yeh

**Affiliations:** 1Department of Orthopedics, Hualien Tzu Chi Hospital, Buddhist Tzu Chi Medical Foundation, Hualien 970473, Taiwan; james800411@tzuchi.com.tw (G.-Y.C.); timwu@tzuchi.com.tw (W.-T.W.); ihchen@tzuchi.com.tw (I.-H.C.); feyu@tzuchi.com.tw (T.-C.Y.); 2School of Medicine, Tzu Chi University, Hualien 970374, Taiwan; 3Institute of Medical Sciences, Tzu Chi University, Hualien 970374, Taiwan; fish@gms.tcu.edu.tw; 4Department of Medical Research, Hualien Tzu Chi Hospital, Buddhist Tzu Chi Medical Foundation, Hualien 970473, Taiwan; paulwang@tzuchi.com.tw; 5Graduate Institute of Clinical Pharmacy, Tzu Chi University, Hualien 970374, Taiwan

**Keywords:** gastrointestinal hemorrhage, perioperative care, fracture osteosynthesis, aged, chronic renal insufficiency

## Abstract

(1) Background: Upper gastrointestinal bleeding (UGIB), a major postoperative complication after surgical fixation of major limb fractures, can be fatal but is often neglected. This study determined the incidence rates of and related risk factors for perioperative UGIB among older patients with major upper limb fractures but without a history of peptic ulcer disease (PUD). (2) Methods: We collected the data of patients aged more than 65 years who underwent surgery for major limb fracture between 1 January 2001 and 31 December 2017, from Taiwan’s National Health Insurance Research Database and excluded those with a history of UGIB and PUD before the date of surgery. The primary outcome was the incidence of UGIB requiring panendoscopy during hospitalization. A multiple logistic regression model was used to identify the independent predictors of UGIB, with adjustment for confounding factors. The final model included variables that were either statistically significant in univariate analyses or deemed clinically important. (3) Results: The incidence of UGIB was 2.8% among patients with major limb fractures. Male sex, older age, major lower limb fracture, and a history of chronic renal disease were significant risk factors for the increased incidence of perioperative UGIB. (4) Conclusions: Patients with major limb fractures who underwent surgery exhibited a higher rate of stress ulceration with UGIB, even when they had no history of PUD. Perioperative preventive protocols (e.g., protocols for the administration of proton-pump inhibitors) may be necessary for patients with these major risk factors.

## 1. Introduction

With the gradual aging of populations worldwide, fragility fractures have evolved into a significant public health issue, resulting in substantial morbidity, mortality, and health-care costs [1,2,3,4]. Osteoporosis, a prevalent condition affecting more than 200 million individuals globally, is the underlying etiology of fragility fractures, leading to approximately 8.9 million fractures annually [5]. The mortality risk associated with fragility fractures is high, with 1-year mortality rates ranging from 5.1% to 12.5% in women and 6.0% to 19.5% in men [6]. In addition to immediate orthopedic concerns, perioperative complications cause further challenges for the treatment of fragility fractures. Among these complications, upper gastrointestinal bleeding (UGIB) has emerged as a salient issue because of its substantial morbidity and mortality [3,7,8]. This complication has a broad range of incidence rates, from 2.5% to 11%, and its mortality can reach 50% [9,10,11,12,13,14]. Although UGIB represents a significant concern, the existing body of literature has inadequately investigated its incidence and the associated risk factors in older adults undergoing surgery for limb fractures [15,16]. This gap in the research indicates the requirement of the present study. Achieving a deeper understanding of the incidence of and risk factors for UGIB in this demographic is critical for several reasons. First, the risk factors for UGIB among elderly persons may differ substantially from those in younger cohorts because of comorbidities and physiological changes associated with aging [17,18,19]. Second, limb fractures themselves may be an independent risk factor that exacerbates pre-existing conditions, thus increasing the likelihood of UGIB [20,21,22]. Lastly, delineating these risk factors is integral to developing targeted prevention strategies, such as optimized protocols for the administration of proton-pump inhibitors or anticoagulants, which may substantially mitigate UGIB risk [23,24,25].

This study thus aimed to fill the existing knowledge gap by investigating the incidence of acute perioperative UGIB and its associated risk factors in a sample of elderly patients who had undergone surgical fixation for major upper limb or lower limb fractures. The findings from this study provide crucial information for both clinical practice and future research, thereby serving to potentially reduce the incidence and associated mortality of UGIB in this vulnerable population.

## 2. Materials and Methods

The present study, approved by the Research Ethics Committee of Hualien Tzu Chi Hospital (IRB 108-242-C), investigated the incidence and associated factors of postoperative UGIB in patients aged more than 65 years who had undergone open reduction and internal fixation (ORIF) surgery for major upper limb (humerus or forearm) or major lower limb (femur or tibia) fractures between 1 January 2001 and 31 December 2017. These patients were identified from Taiwan’s National Health Insurance Research Database (NHIRD) and Longitudinal Health Insurance Database 2000 (LHID2000), which collectively contain the data of approximately 99% of the 23.74 million residents enrolled into the National Health Insurance program in Taiwan, which was, initiated on 1 March 1995. The NHIRD is a nationwide database that contains records coded in accordance with the International Classification of Diseases, Ninth Revision, Clinical Modification (ICD-9-CM) and ICD-10. In addition, the LHID2000, a subset of the NHIRD, contains the data of 2 million beneficiaries sampled at random in 2000, and the data contained in the LHID2000 further augmented the present study’s data. Notably, validation studies of the NHIRD and LHID2000 have been conducted by National Health Research Institute, which have confirmed their representativeness [26].

The inclusion criteria of the study were as follows: patients who were aged more than 65 years and had undergone ORIF surgery for major upper limb fractures (humerus or forearm fracture; surgery code 64239B or 64032B) or major lower limb fractures (femur or tibia fracture; surgery code 64028B, 64029B, or 64031B) within the specified time frame (Figure 1). To establish a consistent cohort, patients with a history of UGIB (ICD-9-CM code 578.9; ICD-10 code K92.2) and peptic ulcer disease (PUD; ICD-9-CM codes 531–533; ICD-10 codes K25–K27, K31, and K56) before the index date (the date of ORIF for multiple trauma or multiple-site fractures; ICD-9-CM code 995.99; ICD-10 code T07) were excluded from the analysis. The primary outcome under investigation was the incidence of UGIB requiring panendoscopy (ICD-9-CM code 578.9; ICD-10 codes K92.2 and 28016C) during hospitalization for ORIF surgery (Figure 1). The researchers also identified demographic factors and comorbidities that are potentially associated with UGIB. These factors include sex, age, and hypertension (ICD-9-CM codes 401–405; ICD-10 codes I10–I13 and I15), diabetes mellitus (DM; ICD-9-CM code 250; ICD-10 codes E10, E11, and E13), dyslipidemia (ICD-9-CM codes 272.0–272.9; ICD-10 code E78), psychiatric disorders (ICD-9-CM codes 290–319; ICD-10 codes F01–F48, F50–F69, and F99), coronary artery disease (ICD-9-CM codes 410–414; ICD-10 codes I20–I25), cerebrovascular accident (ICD-9-CM codes 430–438; ICD-10 codes I60–I69, G45, and G46), peripheral vascular disease (ICD-9-CM codes 443.0–443.9; ICD-10 codes I73), chronic liver disease (ICD-9-CM codes 571.0–571.9; ICD-10 codes K70, K73, K74, K75.4, K76.0, K76.9, K75.81, and K76.89), chronic renal failure (ICD-9-CM codes 585.0–585.9; ICD-10 codes N18.4, N18.5, N18.6, and N18.9), and osteoporosis (ICD-9-CM code 733; ICD-10 codes M80–M82).

Statistical analyses were conducted using SAS version 9.4 (SAS Institute, Cary, NC, USA) and Stata 16.1 (Stata, College Station, TX, USA). Continuous variables are expressed as means and standard deviations, and categorical variables are presented as numbers (i.e., case numbers) and percentages. To identify risk factors for UGIB, data were evaluated using simple and multiple logistic regression analyses. Simple logistic regression was employed to individually assess the impact of demographic factors and comorbidities (e.g., sex, age, hypertension, and DM) on the incidence of postoperative UGIB. In the multiple logistic regression model, we included variables that were statistically significant in the simple logistic regression analysis or were deemed clinically relevant. Further, to rule out the effects of anticoagulants on UGIB incidence, simple and multiple logistic regression subgroup analyses of patients without a history of coronary artery disease, cerebrovascular accident, or peripheral vascular disease were conducted. Statistical significance was defined as *p* values of less than 0.05. Regression coefficients, odds ratios, and their corresponding 95% confidence intervals were calculated. The findings are discussed not just in terms of statistical significance (*p* < 0.05) but also in terms of their clinical relevance. For instance, variables with high odds ratios may be critical risk factors that warrant urgent clinical intervention. The comprehensive methodology employed in this study elucidated the incidence of and potential risk factors for postoperative UGIB in elderly patients with major upper or lower limb fractures undergoing ORIF surgery. Utilizing a large-scale nationwide database ensured robust and representative data, providing valuable insights into the impact of this major complication on postoperative outcomes in this vulnerable population. The research findings contribute to the development of effective preventive measures and improved management strategies for reducing the burden of UGIB in elderly patients undergoing ORIF surgery for limb fractures.

## 3. Results

In total, 18,178 patients were included in the present study; they comprised 6162 men and 12,016 women, who had a mean age of 77.6 ± 7.5 years. Among the patients, 6744 (37.1%) were aged between 65 and 74 years, 7724 (42.5%) were aged between 75 and 84 years, and 3710 (20.4%) were older than 85 years. In total, 6759 (37.2%) of the patients underwent ORIF surgery for major upper limb fractures, and 11,419 (62.8%) underwent ORIF surgery for major lower limb fractures. Regarding comorbidities, 9656 (53.1%) of the patients had hypertension, 4856 (26.7%) had DM, 3065 (16.9%) had dyslipidemia, 2762 (15.2%) had coronary artery disease, 3060 (16.8%) had cerebrovascular accident, 8089 (4.4%) had chronic liver disease, 817 (4.5%) had chronic renal disease, 4083 (22.5%) had psychiatric disorders, 1657 (9.1%) had osteoporosis, and 1133 (15.7%) had peripheral vascular disease (Table 1). Among the included patients, 517 (2.8%) developed perioperative UGIB requiring panendoscopy (Table 1). The risk factors significantly associated with perioperative UGIB requiring panendoscopy were older age (OR, 1.02; 95% CI, 1.00, 1.03; *p* = 0.008), male sex (OR, 1.46; 95% CI, 1.21, 1.75; *p* < 0.001), major lower limb fracture (OR, 2.65; 95% CI, 2.06, 3.41; *p* < 0.001), and chronic renal disease (OR, 2.12; 95% CI, 1.56, 2.89; *p* < 0.001) (Table 2). For the patients without coronary artery disease, cerebrovascular accident, or peripheral vascular disease, the subgroup risk factors significantly associated with perioperative UGIB requiring panendoscopy were older age (OR, 1.02; 95% CI, 1.00, 1.03; *p* = 0.017), male sex (OR, 1.52; 95% CI, 1.21, 1.92; *p* < 0.001), major lower limb fracture (OR, 2.91; 95% CI, 2.11, 4.00; *p* < 0.001), and chronic renal disease (OR, 2.49; 95% CI, 1.63, 3.81; *p* < 0.001) (Table 3).

## 4. Discussion

In this extensive population-based study, we observed an incidence of perioperative UGIB requiring panendoscopy that reached 2.8% among patients aged more than 65 years who underwent ORIF surgery for major limb fractures, even in the absence of a history of PUD and UGIB. UGIB is a known potential complication of surgery, especially in patients with a history of PUD. Studies have highlighted that UGIB is more prevalent in patients with a history of PUD, with older age, and administered nonsteroidal anti-inflammatory drugs perioperatively [17,18]. A 2018 study investigating the incidence of UGIB in patients with hip fracture who underwent surgery reported a very low incidence of perioperative acute UGIB, and preexisting PUD was identified as a risk factor. The study suggested timely endoscopic evaluation and the administration of additional precautions such as gastroprotective agents and nutritional support to optimize patient outcomes [19]. By contrast, our study focused on an older patient population without PUD history, revealing a significant UGIB rate among patients with limb fractures, even in the absence of this risk factor. Clinicians should consider these factors during preoperative evaluations to make appropriate decisions regarding perioperative management.

The primary risk factors identified in our study for UGIB post ORIF surgeries in elderly patients were older age, male sex, lower limb fractures, and comorbidities. In older patients, chronic illnesses and comorbidities increase the likelihood of stress ulceration. A recent nationwide population-based study conducted in 2021 analyzed postoperative gastrointestinal bleeding and its associated risk factors, revealing that major gastrointestinal and cardiovascular surgeries were associated with high rates of postoperative gastrointestinal bleeding. Moreover, older age, male sex, lower income, comorbidities, PUD, congestive heart failure, and steroid use were all linked to an increased risk of postoperative gastrointestinal bleeding [20]. Previous studies have also reported that bedridden patients with lower limb fractures exhibited a higher incidence of pneumonia within 30 days of hip fracture surgery, which further contributes to an increased risk of UGIB and increased mortality rates [21,22,23]. Given these factors, clinicians must pay close attention to this perioperative complication while providing postoperative care and administering medication to patients with lower limb fractures.

Through multivariate logistic analysis, we further discovered that chronic renal disease is a significant risk factor for postoperative UGIB. Studies have indicated that patients with chronic renal disease have an increased risk of perioperative peptic ulcer bleeding, bleeding-related morbidity, and mortality [24,25,27,28]. The mechanisms underlying these increased risks necessitate further investigation. The association between chronic renal disease and platelet dysfunction may lead to impaired hemostasis [29]. Another plausible mechanism is the association of chronic renal failure with increased levels of proinflammatory cytokines and oxidative stress, resulting in tissue damage and increased susceptibility of the gastrointestinal mucosa to injury [30]. Such mechanisms may contribute to the increased incidence of perioperative UGIB in patients with chronic renal disease. Clinicians caring for older patients who have undergone ORIF surgery for limb fractures must recognize the potential associations between chronic renal disease and UGIB as well as the interaction between nonsteroidal anti-inflammatory drug use and UGIB risk. Consequently, preventive measures for UGIB should be implemented for older patients with chronic renal disease, regardless of the painkiller type or peptic ulcer history, to enhance postoperative care quality and patient outcomes. Our extensive study involving a large patient population aged more than 65 years undergoing ORIF surgery for major limb fractures found a significant incidence of perioperative UGIB requiring panendoscopy, even in the absence of a history of PUD and UGIB. This finding highlights the importance of careful preoperative patient evaluation and individual risk factor assessment for UGIB during perioperative management decision-making. We identified older age, male sex, lower limb fractures, and chronic renal disease as risk factors for postoperative UGIB, which should guide clinicians in providing vigilant postoperative care for these patients and implementing appropriate preventive measures to reduce the burden of UGIB and improve patient outcomes. Further research is warranted to explore the mechanisms underlying the associations between these risk factors and UGIB, with the ultimate goal of enhancing perioperative care strategies for older patients undergoing ORIF surgery for limb fractures.

The correlation between limb fractures and perioperative UGIB among elderly individuals has been a topic of interest in the related literature. Various studies have investigated this association to gain a better understanding of the risk factors and potential preventive measures for this complication in this vulnerable population [31]. Elderly individuals, especially those aged more than 65 years, are at increased risks of both limb fractures and UGIB due to age-related changes in the musculoskeletal system and gastrointestinal tract [32]. Fragility fractures, such as those affecting the hip, femur, humerus, and forearm, are common in the elderly population and are mainly attributed to decreased bone density and increased frailty. These fractures often require surgical intervention, such as ORIF, to promote healing and functional recovery. Several studies have highlighted the notable incidence of perioperative UGIB in elderly patients with limb fractures who have undergone ORIF. The incidence of UGIB in this population can range from 2.8% to 11%, depending on the specific fracture type, patient demographics, and comorbidities [1,2,3,4,5,6]. Older age, male sex, and lower limb fractures have consistently emerged as significant risk factors for UGIB among these patients. One potential mechanism contributing to the increased risk of UGIB in elderly individuals with limb fractures is stress ulceration. Fractures and their surgical treatment can lead to significant physiological stress, resulting in mucosal damage and increased vulnerability of the gastrointestinal tract to bleeding [33]. Proton-pump inhibitors have been established as a first-line treatment and prophylaxis against stress ulcers and related UGIB, particularly in high-risk patients [34]. They irreversibly inhibit the hydrogen–potassium ATPase pump, thereby reducing gastric acid secretions and creating a more favorable environment for mucosal healing [35]. Our study demonstrated that the utilization of proton-pump inhibitors can mitigate the risk of postoperative UGIB, particularly in patients with identified risk factors. Further, prolonged bed rest and immobilization after lower limb fractures may contribute to venous stasis and increase the risk of deep vein thrombosis, which may further exacerbate the risk of UGIB [36]. Postsurgical immobility may contribute to venous stasis, which is a well-known risk factor for deep vein thrombosis [37]. In a state of venous stasis, blood flow is slow, increasing the risk of clot formation. Once formed, clots can dislodge and cause embolic events, further exacerbating the risk of UGIB owing to the potential requirement for anticoagulant therapy, which itself is a gastrointestinal bleeding risk [38]. Further, comorbidities and polypharmacy are common in the elderly population, which can further enhance the risk of UGIB. Conditions such as PUD, chronic renal failure, and cardiovascular disease and the use of nonsteroidal anti-inflammatory drugs are recognized risk factors for UGIB and are often prevalent in older adults with limb fractures [5,6]. In summary, the literature supports a correlation between limb fractures and perioperative UGIB development in the elderly population, particularly in those aged more than 65 years. To mitigate the risk of perioperative UGIB in elderly patients with limb fractures, several preventive measures are recommended. Timely endoscopic evaluation may be vital to identify and treat any preexisting peptic ulcers or gastritis before surgery. Further, nonsteroidal anti-inflammatory drug use should be minimized, or alternative pain management strategies should be adopted to reduce the risk of potential gastrointestinal complications.

Older age, male sex, lower limb fractures, and comorbidities were demonstrated to play significant roles in increasing the risk of UGIB in our nationwide-based cohort study. By recognizing these risk factors and implementing appropriate preventive measures, clinicians can improve perioperative care for elderly patients with limb fractures and reduce the incidence of UGIB, ultimately leading to better postoperative outcomes and overall patient well-being. The correlation between ORIF surgeries for limb fractures and perioperative UGIB in the elderly group has been a subject of interest in medical research. As the elderly population is at increased risks of both limb fractures and UGIB, understanding the potential association between these two factors is crucial for improving perioperative care and patient outcomes [8,9,10,11]. The surgical procedure itself can contribute to the risk of UGIB in elderly patients. ORIF surgeries are complex and may lead to significant physiological stress during the operation, which can lead to mucosal damage in the gastrointestinal tract. The use of anesthesia and postoperative analgesics also affects the gastrointestinal system and contribute to the risk of bleeding [39,40]. Moreover, immobilization and bed rest after ORIF surgeries, especially in the case of lower limb fractures, can increase the risk of venous stasis and deep vein thrombosis, further predisposing patients to UGIB [36]. During the postoperative period, close monitoring of patients for signs of UGIB is critical because early detection can lead to timely intervention and better outcomes [10,11]. Proton-pump inhibitors or other gastroprotective agents may be prescribed to reduce the risk of stress-related mucosal damage in high-risk patients [41,42]. A notable correlation between ORIF surgeries for limb fractures and perioperative UGIB has been observed in the elderly population. The complexity of the surgical procedure, physiological stress, immobilization, comorbidities, and the use of nonsteroidal anti-inflammatory drugs all contribute to an increased risk of UGIB in this patient group [36,43]. By understanding these risk factors and implementing appropriate preventive measures, clinicians can optimize perioperative care for elderly patients undergoing ORIF surgeries for limb fractures and reduce the incidence of UGIB, ultimately leading to improved postoperative outcomes and better quality of life for these patients.

Implementing effective preventive measures based on the identified risk factors may lead to a considerable reduction in health-care costs. Moreover, early detection and prevention would significantly enhance patient quality of life by reducing morbidity and potential mortality associated with postoperative UGIB [44]. However, introducing preventive measures may pose ethical dilemmas related to patient autonomy and informed consent, particularly in elderly populations that may already be taking multiple medications [45]. Our findings demonstrated a significant correlation between ORIF surgeries for limb fractures and perioperative UGIB in the elderly population. Our investigation revealed several key risk factors for UGIB, including the complexity of the surgical procedure, physiological stress, immobilization, pre-existing comorbidities, and the use of nonsteroidal anti-inflammatory drugs.

The present study has several limitations. First, this was a retrospective cohort study that focused on a highly specific population, namely patients with major limb fractures and postoperative UGIB. Consequently, potential selection bias exists. Specifically, patients with these distinct characteristics may exhibit certain physiological or medical conditions that are not universally applicable. Thus, although this study provides valuable insights into this particular subgroup, caution is warranted when extrapolating the findings to a heterogeneous patient population. Second, this study did not consider perioperative doses of anticoagulant or nonsteroidal anti-inflammatory drugs. The presence or absence of these medications, however, is a significant confounding factor, given their established role as a risk factor for UGIB. Although we performed a subgroup analysis that excluded patients with cardiovascular comorbidities—who are more likely to be on anticoagulant medications—residual confounding effects may have persisted, which, in turn, may have profoundly affected our analytic results. Third, the database used in this study lacks specific data on gastrointestinal pathologies and *H. pylori* infection, which play pivotal roles in both the development and treatment of peptic ulcers and may serve as a crucial etiological factor for UGIB. The absence of such data significantly limits the depth of our analysis and the comprehensiveness of our findings. For example, patients with an inlet patch frequently have Barrett’s esophagus, which frequently involves *H. pylori* colonization and may predispose the individual to gastroesophageal reflux, causing further peptic ulcers [46]. Fourth, another salient limitation is the absence of data on patients’ postoperative pain levels, intake conditions, and activity states, which may contribute to the development of UGIB. The level of postoperative discomfort, nutritional intake, and physical activity can affect gastrointestinal motility and acidity, thus serving as potential risk or protective factors for gastrointestinal complications. Despite the aforementioned limitations, the present study serves as a seminal investigation into the significance of UGIB prevention among older adults with major limb fractures. This study identified key risk factors, besides PUD, that are highly correlated with this postoperative complication. The findings provide a basis for the implementation of targeted prevention protocols, such as administering proton-pump inhibitors and antacids, encouraging dietary adjustments, providing effective pain control, and recommending activity modification; however, certain inherent challenges and complexities are associated with the development and execution of these measures.

## 5. Conclusions

The present retrospective cohort study, based on a nationwide population database, indicates that patients aged more than 65 years with major limb fractures, particularly lower limb fractures, are at an elevated risk of perioperative UGIB, even in the absence of history of PUD. This study identified older age, male sex, lower limb fracture, and chronic renal disease as salient risk factors for such complications. Given the potentially devastating and fatal outcomes of stress ulceration with UGIB—including substantial mortality rates, organ failure, and extended hospitalization—targeted preventive measures are necessary. These may involve endoscopic evaluation, the administration of gastroprotective agents such as proton-pump inhibitors, the encouragement of dietary adjustments, and the implementation of effective pain control strategies. In this regard, the findings of this study can inform the development of evidence-based guidelines for perioperative care that target this vulnerable demographic. Moreover, this study sets the stage for broader discussions within health-care systems and policy-making frameworks by emphasizing the need for a multidisciplinary approach for managing the perioperative risks of elderly patients with major limb fractures. For future research directions, these findings should be validated through prospective studies, and the cost-effectiveness and patient outcomes associated with different prevention protocols should be evaluated. Further, the implementation and assessment of these protocols in a real-world clinical setting would invaluable insights, enhancing the body of knowledge in this domain.

## Figures and Tables

**Figure 1 healthcare-11-02853-f001:**
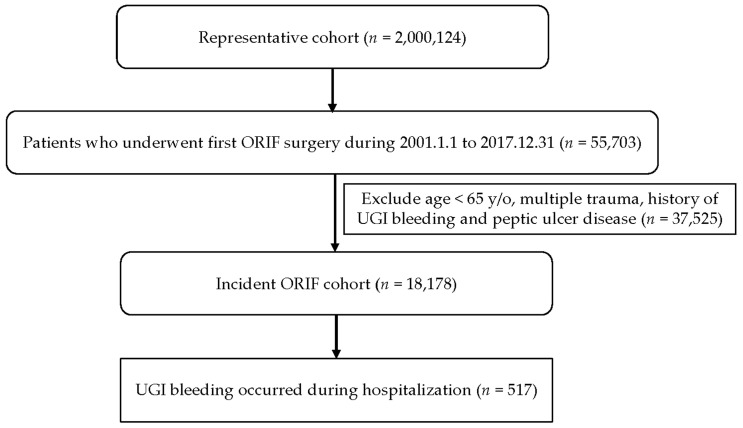
Study design and case inclusion.

**Table 1 healthcare-11-02853-t001:** Demographic data of patients with single major limb fracture receiving ORIF surgery (*n* = 18,178).

Variables	Male	Female	Total
N	6162	12,016	18,178
Age	77.9 ± 7.5	77.4 ± 7.6	77.6 ± 7.5
Age Group	-	-	-
65–74 y/o	2146 (34.8%)	4598 (38.2%)	6744 (37.1%)
75–84 y/o	2705 (43.9%)	5019 (41.8%)	7724 (42.5%)
≥85 y/o	1311 (21.3%)	2399 (20.0%)	3710 (20.4%)
Fracture site	-	-	-
Upper major limb	1484 (24.1%)	5275 (43.9%)	6759 (37.2%)
Lower major limb	4678 (75.9%)	6741 (56.1%)	11,419 (62.8%)
Hypertension (%)	2929 (47.5%)	6727 (56.0%)	9656 (53.1%)
Diabetes mellitus (%)	1288 (20.9%)	3568 (29.7%)	4856 (26.7%)
Dyslipidemia (%)	703 (11.4%)	2362 (19.7%)	3065 (16.9%)
Coronary artery disease (%)	980 (15.9%)	1782 (14.8%)	2762 (15.2%)
Cerebrovascular accident (%)	1217 (19.8%)	1843 (15.3%)	3060 (16.8%)
Chronic liver disease (%)	260 (4.2%)	548 (4.6%)	808 (4.4%)
Chronic renal disease (%)	356 (5.8%)	461 (3.8%)	817 (4.5%)
Psychiatric disorders (%)	1217 (19.8%)	2866 (23.9%)	4083 (22.5%)
Osteoporosis (%)	300 (4.9%)	1357 (11.3%)	1657 (9.1%)
Peripheral vascular disease (%)	433 (7.0%)	700 (5.8%)	1133 (6.2%)
Upper gastrointestinal bleeding (%)	241 (3.9%)	276 (2.3%)	517 (2.8%)

ORIF: Open reduction and internal fixation. Data are presented as *n* or mean ± standard deviation.

**Table 2 healthcare-11-02853-t002:** Risk factors associated with UGIB among patients with single major limb fracture receiving ORIF surgery (*n* = 18,178).

Variables	Crude		Adjusted	
	Odds Ratio (95% CI)	*p* Value	Odds Ratio (95% CI)	*p* Value
Age	1.03 (1.02, 1.05)	<0.001 *	1.02 (1.00, 1.03)	0.008 *
Gender (Male vs. Female)	1.73 (1.45, 2.06)	<0.001 *	1.46 (1.21, 1.75)	<0.001 *
Fracture site (Lower major limb vs. Upper major limb)	3.18 (2.51, 4.03)	<0.001 *	2.65 (2.06, 3.41)	<0.001 *
Hypertension vs. None	1.04 (0.87, 1.23)	0.696	1.01 (0.84, 1.23)	0.879
Diabetes mellitus vs. None	1.02 (0.84, 1.24)	0.849	1.01 (0.81, 1.25)	0.959
Dyslipidemia vs. None	0.85 (0.66, 1.08)	0.184	0.98 (0.75, 1.27)	0.875
Coronary artery disease vs. None	1.13 (0.90, 1.43)	0.294	1.02 (0.80, 1.30)	0.897
Cerebrovascular accident vs. None	1.06 (0.84, 1.33)	0.636	0.92 (0.72, 1.16)	0.479
Chronic liver disease vs. None	1.24 (0.84, 1.83)	0.278	1.32 (0.89, 1.96)	0.167
Chronic renal disease vs. None	2.41 (1.79, 3.25)	<0.001 *	2.12 (1.56, 2.89)	<0.001 *
Psychiatric disorders vs. None	0.99 (0.81, 1.23)	0.989	0.98 (0.79, 1.21)	0.826
Osteoporosis vs. None	0.95 (0.70, 1.29)	0.742	0.99 (0.73, 1.36)	0.984
Peripheral vascular disease vs. None	1.24 (0.89, 1.72)	0.212	1.06 (0.75, 1.48)	0.745

ORIF: Open reduction and internal fixation. Data are presented as Odds ratio (95% CI). * *p* < 0.05 was considered statistically significant after test.

**Table 3 healthcare-11-02853-t003:** Subgroup risk factors associated with UGIB among patients with single major limb fracture receiving ORIF surgery and without coronary artery disease, cerebrovascular accident, or peripheral vascular disease (*n* = 12,262).

Variables	Crude		Adjusted	
	Odds Ratio (95% CI)	*p* Value	Odds Ratio (95% CI)	*p* Value
Age	1.04 (1.03, 1.05)	<0.001 *	1.02 (1.00, 1.03)	0.017 *
Gender (Male vs. Female)	1.87 (1.50, 2.34)	<0.001 *	1.52 (1.21, 1.92)	<0.001 *
Fracture site (Lower major limb vs. Upper major limb)	3.59 (2.67, 4.85)	<0.001 *	2.91 (2.11, 4.00)	<0.001 *
Hypertension vs. None	0.98 (0.78, 1.22)	0.847	1.02 (0.80, 1.29)	0.902
Diabetes mellitus vs. None	0.88 (0.68, 1.16)	0.370	0.89 (0.66, 1.20)	0.432
Dyslipidemia vs. None	0.72 (0.50, 1.02)	0.068	0.92 (0.63, 1.34)	0.655
Chronic liver disease vs. None	1.30 (0.79, 2.13)	0.305	1.48 (0.89, 2.45)	0.127
Chronic renal disease vs. None	2.72 (1.80, 4.12)	<0.001 *	2.49 (1.63, 3.81)	<0.001 *
Psychiatric disorders vs. None	1.03 (0.78, 1.36)	0.846	1.02 (0.76, 1.35)	0.915
Osteoporosis vs. None	0.87 (0.57, 1.33)	0.525	0.91 (0.60, 1.40)	0.676

ORIF: Open reduction and internal fixation. Data are presented as odds ratio (95% CI). * *p* < 0.05 was considered statistically significant after test.

## Data Availability

Data are contained within the article.

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
