# Peer review of "Incidence of Acute Upper Gastrointestinal Bleeding and Related Risk Factors among Elderly Patients Undergoing Surgery for Major Limb Fractures: An Analytical Cohort Study"

_healthcare, 2023, doi:10.3390/healthcare11212853_

Round 1
Reviewer 1 Report
Comments and Suggestions for Authors
Dear authors,
I appreciate your submission of your study to this journal:
Your study provides a comprehensive assessment of the risk factors associated with acute upper gastrointestinal bleeding in older patients undergoing surgery for major limb fractures.
this is my review
Abstract : You need to specify in brief the kind of statistical analysis
Introduction:
the introduction outlines the importance of the study and the context for investigating perioperative UGIB in older adults with fragility fractures mentioning "limited research" on the topic, but it's not clear how the significance of the topic was determined. More context on why this research is needed and the potential impact of the findings would enhance the introduction.
M & M:
This part is well written, I have few considerations:
Model Specification: The section doesn't go into detail about the specific variables included in the regression models (simple and multiple), which could affect the study's results and interpretation. Limitations of Subgroup Analysis: While subgroup analyses can be useful, they may have limitations if the sample size within subgroups becomes small, affecting the reliability of the results.
Interpretation of Results: It's crucial to provide guidance on how the results of the regression analyses will be interpreted, especially in terms of clinical significance.
Validation and Robustness: While the study mentions that using a large-scale nationwide database ensures robust and representative data, it would be beneficial to provide additional information about data validation or measures taken to address potential biases.
Overall, the section effectively outlines the process of statistical analysis, covering the use of appropriate software, variable presentation, regression analysis, and subgroup analyses. To fully assess the quality and reliability of the analysis, it is important to have the opportunuty to examine the complete methodology of the study, including model specifications, providing potential covariates, and all other statistical details that could influence the results
Results:
Ok
Discussion:
The section present a comprehensive analysis of the study's findings and contextualizes them within existing research
I have few suggestion to improve the quality of this section:
- The mention of proton pump inhibitors or other gastroprotective agents is a valuable addition. You can expand on their role and effectiveness in reducing the risk of stress-related mucosal damage.
- While you discuss immobilization and bed rest as factors increasing the risk of venous stasis, you could briefly mention how these factors may lead to an increased risk of deep vein thrombosis, elaborating on the mechanism if necessary.
- Consider providing a short summary of the main risk factors identified in your study (older age, male sex, lower limb fractures, comorbidities) before moving into the broader discussion of these factors' implications.
- To emphasize the significance of the study's findings, you might mention the potential healthcare cost reduction and improvement in patient quality of life that could result from effective preventive measures.
- If there are any ethical or practical implications related to implementing the recommended preventive measures, it could be worth mentioning them briefly.
- Consider rephrasing or reorganizing the sentences that mention "there may be a notable correlation" to create a smoother flow in the discussion.
- For clarity, you could provide a brief overview of the key risk factors (e.g., complex surgical procedure, physiological stress, immobilization, comorbidities, drug use) before delving into the discussion of each factor's contribution to upper gastrointestinal bleeding.
- As you discuss the study's limitations, consider expanding on the concept of selection biases related to the patients and how these biases might impact the generalizability of the results.
- When mentioning the potential influence of anticoagulant medications, you could further emphasize the significance of this limitation and its potential impact on the study's findings.
- In discussing the absence of H. pylori infection data, you might briefly mention its relevance to the development and treatment of peptic ulcers.
- When discussing pain levels, intake conditions, and activity conditions, consider explaining how these factors might influence upper gastrointestinal bleeding, even if you don't have specific data on them.
- You could elaborate on the potential challenges and complexities associated with developing and implementing prevention protocols, such as dietary adjustments, effective pain control, and activity modification.
Conclusion
good, I have few observations
- Consider providing a brief recap of the preventive measures you mentioned earlier in the discussion section (such as endoscopic evaluation, gastroprotective agents, dietary adjustments, and pain control). This would provide a comprehensive summary of potential strategies.
- While mentioning the "devastating and fatal complications," you could briefly mention some of these complications to highlight the gravity of the situation, such as potential mortality rates or severe health consequences or delete this statement.
- To strengthen the conclusion, consider including a forward-looking statement. For instance, you could suggest that the findings of this study may help inform the development of evidence-based guidelines for perioperative care in elderly patients with major limb fractures.
- If relevant, you might touch on the broader implications of your findings for healthcare systems, policy-making, or future research directions.
The suggestions are intended to enhance the clarity and impact of your conclusions, and you should tailor them to fit the specific content and context of your study.
Sincerly
Author Response
Comments and Suggestions for Authors
Dear authors,
I appreciate your submission of your study to this journal:
Your study provides a comprehensive assessment of the risk factors associated with acute upper gastrointestinal bleeding in older patients undergoing surgery for major limb fractures.
this is my review
Abstract : You need to specify in brief the kind of statistical analysis
Ans: Thank you for your suggestions. I have added the below description as below:” A multiple logistic regression model was developed to identify independent predictors of UGIB, adjusting for confounding factors by . The final model included variables that were either statistically significant in univariate analyses or deemed clinically important”
Introduction:
the introduction outlines the importance of the study and the context for investigating perioperative UGIB in older adults with fragility fractures mentioning "limited research" on the topic, but it's not clear how the significance of the topic was determined. More context on why this research is needed and the potential impact of the findings would enhance the introduction.
Ans: Thank you for your suggestions. We have revised our introduction as below:” Among these complications, upper gastrointestinal bleeding (UGIB) has emerged as a salient issue because of its substantial morbidity and mortality [3,7-8]. This complication has a broad range of incidence rates, from 2.5% to 11%, and its mortality can reach 50% [9-14]. Although UGIB represents a significant concern, the existing body of literature has inadequately investigated its incidence and the associated risk factors in older adults undergoing surgery for limb fractures [15,16]. This gap in the research indicates the requirement of the present study. Achieving a deeper understanding of the incidence of and risk factors for UGIB in this demographic is critical for several reasons. First, the risk factors for UGIB among elderly persons may differ substantially from those in younger cohorts because of comorbidities and physiological changes associated with aging [17-19]. Second, limb fractures themselves may be an independent risk factor that exacerbates pre-existing conditions, thus increasing the likelihood of UGIB [20-22]. Lastly, delineating these risk factors is integral to developing targeted prevention strategies, such as optimized protocols for the administration of proton pump inhibitors or anticoagulants, which may substantially mitigate UGIB risk [23-25].
This study thus aimed to fill the existing knowledge gap by investigating the incidence of acute perioperative UGIB and its associated risk factors in a sample of elderly patients who had undergone surgical fixation for major upper limb or lower limb fractures. The findings from this study provide crucial information for both clinical practice and future research, thereby serving to potentially reduce the incidence and associated mortality of UGIB in this vulnerable population.”
M & M:
This part is well written, I have few considerations:
Model Specification: The section doesn't go into detail about the specific variables included in the regression models (simple and multiple), which could affect the study's results and interpretation.
Ans: This section has been supplemented as below:” Simple logistic regression was employed to individually assess the impact of demographic factors and comorbidities (e.g., sex, age, hypertension, diabetes mellitus, etc.) on the incidence of postoperative upper gastrointestinal bleeding. In the multiple logistic regression model, we included variables that were statistically significant in the simple logistic regression analysis or were deemed clinically relevant.”
Limitations of Subgroup Analysis: While subgroup analyses can be useful, they may have limitations if the sample size within subgroups becomes small, affecting the reliability of the results.
Ans: While subgroup analyses were performed to isolate the effects of anticoagulants on UGIB incidence among patients without a history of specific vascular diseases, it must be noted that the sample size within these subgroups may be limited, potentially affecting the statistical robustness of the findings. According to a previous study regarding the sample size calculation, we needed at least 4643 cases for analysis (k: number of independent variables in the model. p: the minimum value of p1 or p2. p1 represents the proportion of patients with outcome. p2 represents the proportion of patients without outcome. In our study, k=13 and p=2.8%. Thus, the minimum sample size needed is 4,643. We had 12262 cases). Our sample size is sufficient.
Interpretation of Results: It's crucial to provide guidance on how the results of the regression analyses will be interpreted, especially in terms of clinical significance.
Ans: This section has been supplemented as below:” Regression coefficients, odds ratios, and their corresponding 95% confidence intervals will be reported. The findings will be discussed not just in terms of statistical significance (p<0.05) but also in terms of their clinical relevance. For instance, variables with high odds ratios may indicate critical risk factors that warrant urgent clinical intervention.”
Validation and Robustness: While the study mentions that using a large-scale nationwide database ensures robust and representative data, it would be beneficial to provide additional information about data validation or measures taken to address potential biases.
Ans: The National Health Insurance Research Database and Longitudinal Health Insurance Database 2000 have undergone validation studies confirming their representativeness by National Health Research Institute [26].
- National Health Insurance Research Database. Data subsets [cited March 1, 2018]. Available from: https://nhird.nhri.org.tw/en/Data_Subsets.html. Accessed March 29, 2019.
Overall, the section effectively outlines the process of statistical analysis, covering the use of appropriate software, variable presentation, regression analysis, and subgroup analyses. To fully assess the quality and reliability of the analysis, it is important to have the opportunity to examine the complete methodology of the study, including model specifications, providing potential covariates, and all other statistical details that could influence the results
Ans: Thank you for your suggestions. We have modified our material and method section as above replies.
Results: Ok
Discussion:
The section presents a comprehensive analysis of the study's findings and contextualizes them within existing research. I have few suggestions to improve the quality of this section: The mention of proton pump inhibitors or other gastroprotective agents is a valuable addition. You can expand on their role and effectiveness in reducing the risk of stress-related mucosal damage. While you discuss immobilization and bed rest as factors increasing the risk of venous stasis, you could briefly mention how these factors may lead to an increased risk of deep vein thrombosis, elaborating on the mechanism if necessary.
Ans: Thank you for your suggestions. We have added the below content into fourth paragraph of Discussion section as below:” Proton pump inhibitors have been established as a first-line treatment for prophylaxis against stress ulcers and related upper gastrointestinal bleeding, particularly in high-risk patients [34]. They act by irreversibly inhibiting the hydrogen-potassium ATPase pump, thereby reducing gastric acid secretion and creating a more favorable environment for mucosal healing [35]. In our study, the utilization of proton pump inhibitors could potentially mitigate the risk of postoperative upper gastrointestinal bleeding, particularly in patients with identified risk factors. Additionally, prolonged bed rest and immobilization after lower limb fractures may contribute to venous stasis and increase the risk of deep vein thrombosis, which may further exacerbate the risk of upper gastrointestinal bleeding [36]. Immobility post-surgery can contribute to venous stasis, which is a well-known risk factor for the development of deep vein thrombosis [37]. In a state of venous stasis, blood flow is slowed, increasing the risk of clot formation. Once formed, these clots can dislodge and cause embolic events, further exacerbating the risk of upper gastrointestinal bleeding due to the potential requirement for anticoagulant therapy, which itself is a gastrointestinal bleeding risk [38].
Consider providing a short summary of the main risk factors identified in your study (older age, male sex, lower limb fractures, comorbidities) before moving into the broader discussion of these factors' implications.
Ans: Thank you for your suggestions. I have modified this sentence in second paragraph of Discussion as below:” The primary risk factors identified in our study for upper gastrointestinal bleeding post-ORIF surgeries in elderly patients were older age, male sex, lower limb fractures, and the presence of comorbidities.”
To emphasize the significance of the study's findings, you might mention the potential healthcare cost reduction and improvement in patient quality of life that could result from effective preventive measures. If there are any ethical or practical implications related to implementing the recommended preventive measures, it could be worth mentioning them briefly. Consider rephrasing or reorganizing the sentences that mention "there may be a notable correlation" to create a smoother flow in the discussion. For clarity, you could provide a brief overview of the key risk factors (e.g., complex surgical procedure, physiological stress, immobilization, comorbidities, drug use) before delving into the discussion of each factor's contribution to upper gastrointestinal bleeding.
Ans: Thank you for your suggestions. We have added the below content into sixth paragraph of Discussion section as below:” Implementing effective preventive measures based on these identified risk factors could lead to a considerable reduction in healthcare costs. Moreover, early detection and prevention would significantly enhance patient quality of life by reducing morbidity and potential mortality associated with postoperative upper gastrointestinal bleeding [44]. Introducing preventive measures may pose ethical dilemmas related to patient autonomy and informed consent, particularly in elderly populations who may already be taking multiple medications [45]. Our findings demonstrate a significant correlation between ORIF surgeries for limb fractures and subsequent perioperative upper gastrointestinal bleeding in the elderly population. Our investigation unearthed several key risk factors contributing to upper gastrointestinal bleeding, including the complexity of the surgical procedure, physiological stress, immobilization, pre-existing comorbidities, and the use of nonsteroidal anti-inflammatory drugs.”
As you discuss the study's limitations, consider expanding on the concept of selection biases related to the patients and how these biases might impact the generalizability of the results.
Ans: Thank you for your suggestions. We have modified our first limitation in seventh paragraph of Discussion as below:” it is imperative to underscore that this is a retrospective cohort study that centered on a highly specific population, namely patients with major limb fractures and postoperative upper gastrointestinal bleeding. Consequently, the potential for selection bias cannot be overstated. Specifically, patients with these distinct characteristics may exhibit certain physiological or medical conditions that are not universally applicable. Thus, while the study provides valuable insights into this particular subgroup, caution is warranted in extrapolating the findings to a more heterogeneous patient population.”
When mentioning the potential influence of anticoagulant medications, you could further emphasize the significance of this limitation and its potential impact on the study's findings.
Ans: Thank you for your suggestions. We have modified our second limitation in seventh paragraph of Discussion as below:” Second, it is worth emphasizing the study's omission of perioperative doses of anticoagulant or nonsteroidal anti-inflammatory drugs. The presence or absence of these medications is a significant confounding factor, given their established role as a risk factor for upper gastrointestinal bleeding. Although we did perform a subgroup analysis excluding patients with cardiovascular comorbidities who are more likely to be on anticoagulant medications, it is likely that the residual confounding may persist, which, in turn, could have a pronounced impact on our analytic results.”
In discussing the absence of H. pylori infection data, you might briefly mention its relevance to the development and treatment of peptic ulcers.
Ans: Thank you for your suggestions. We have modified our third limitation in seventh paragraph of Discussion as below:” Third, it is notable that the database accessed for this study lacks specific data on gastrointestinal pathologies and H. pylori infection, which are pivotal in both the development and treatment of peptic ulcers and may serve as a crucial etiological factor for upper gastrointestinal bleeding. The absence of such data significantly limits the depth of our analysis and the comprehensiveness of our findings. For example, the patients with inlet patch frequently had Barrett's esophagus which frequently involved H. pylori colonization and may predispose gastroesophageal reflux, causing further peptic ulcer”
When discussing pain levels, intake conditions, and activity conditions, consider explaining how these factors might influence upper gastrointestinal bleeding, even if you don't have specific data on them.
Ans: Thank you for your suggestions. We have modified our fourth limitation in seventh paragraph of Discussion as below:” Fourth, another salient limitation is the absence of data concerning patients' post-operative pain levels, intake conditions, and activity states, which could have a bearing on the development of upper gastrointestinal bleeding. The level of post-operative discomfort, nutritional intake, and physical activity can affect gastrointestinal motility and acidity, thus serving as potential risk or protective factors for gastrointestinal complications.”
You could elaborate on the potential challenges and complexities associated with developing and implementing prevention protocols, such as dietary adjustments, effective pain control, and activity modification.
Ans: Thank you for your suggestions. I have modified our related manuscript section as below:” Despite the aforementioned limitations, the present study serves as a seminal investigation into the significance of upper gastrointestinal bleeding prevention among older adults with major limb fractures. The study identifies key risk factors, aside from peptic ulcer disease, that are highly correlated with this postoperative complication. While the findings advocate for the implementation of targeted prevention protocols—such as administration of proton pump inhibitors and antacids, dietary adjustments, effective pain control, and activity modification—it is crucial to acknowledge the inherent challenges and complexities associated with their development and execution.”
Conclusion: good, I have few observations.
Consider providing a brief recap of the preventive measures you mentioned earlier in the discussion section (such as endoscopic evaluation, gastroprotective agents, dietary adjustments, and pain control). This would provide a comprehensive summary of potential strategies.
While mentioning the "devastating and fatal complications," you could briefly mention some of these complications to highlight the gravity of the situation, such as potential mortality rates or severe health consequences or delete this statement.
To strengthen the conclusion, consider including a forward-looking statement. For instance, you could suggest that the findings of this study may help inform the development of evidence-based guidelines for perioperative care in elderly patients with major limb fractures.
If relevant, you might touch on the broader implications of your findings for healthcare systems, policy-making, or future research directions.
Ans: Thank you for your suggestions. I have modified Conclusion section as below:” In summary, the present retrospective cohort study, based on a nationwide population, underscores those patients aged over 65 years with major limb fractures, particularly lower limb fractures, are at an elevated risk of perioperative upper gastrointestinal bleeding, even in the absence of a prior history of peptic ulcer disease. The study identified older age, male sex, lower limb fracture, and chronic renal disease as salient risk factors for such a complication. Given the potentially devastating and fatal outcomes of stress ulceration with upper gastrointestinal bleeding—including substantial mortality rates, organ failure, and extended hospitalization—targeted preventive measures are indispensable. These may involve endoscopic evaluation, the administration of gastroprotective agents such as proton pump inhibitors, dietary adjustments, and effective pain control strategies. In this regard, the findings of this study could be instrumental in informing the development of evidence-based guidelines for perioperative care, specifically targeting this vulnerable demographic. Moreover, the study sets the stage for broader discussions within healthcare systems and policy-making frameworks, emphasizing the need for a multidisciplinary approach in managing the perioperative risks for elderly patients with major limb fractures. For future research directions, there is a compelling need to validate these findings through prospective studies, and to evaluate the cost-effectiveness and patient outcomes associated with different prevention protocols. Furthermore, the implementation and assessment of these protocols in a real-world clinical setting could offer invaluable insights, furthering the existing body of knowledge in this domain."
The suggestions are intended to enhance the clarity and impact of your conclusions, and you should tailor them to fit the specific content and context of your study.
Ans: Thank you very much for your suggestions. I have modified our manuscript by your suggestions.

Reviewer 2 Report
Comments and Suggestions for Authors
Sep 08, 2023
Dear Author,
1. The manuscript, healthcare-2564972, is within the scope of the journal.
2. The section “Abbreviations” has not been recognized.
3. The section “Keywords” has to be compatible with the MeSH database.
4. In my opinion, the incidence rates and related risk factors of perioperative upper GIS bleeding in elderly with major limb fractures but without a history of PU disease is an important issue. However, the section “Discussion” must be revised, and the reference list must be enriched with several articles, such as “Inlet patch: Associations with endoscopic findings in the upper gastrointestinal system.”
5. The orthographical and grammatical errors have been recognized throughout the text. Of note, the manuscript must be revised by a native English speaker.
6. The manuscript might be accepted after major revision.
Best Regards,
Reviewer, Healthcare

The orthographical and grammatical errors have been recognized throughout the text. Of note, the manuscript must be revised by a native English speaker.
Author Response
Comments and Suggestions for Authors
Dear Author,
- The manuscript, healthcare-2564972, is within the scope of the journal.
Ans: Thank you very much.
- The section “Abbreviations” has not been recognized.
Ans: We have added Abbreviations into our manuscript. Thank you.
- The section “Keywords” has to be compatible with the MeSH database.
Ans: We have added MeSH related keywords as below: “Gastrointestinal Hemorrhage; Perioperative Care; Fracture Osteosynthesis; Aged; Chronic Renal Insufficiency”
- In my opinion, the incidence rates and related risk factors of perioperative upper GIS bleeding in elderly with major limb fractures but without a history of PU disease is an important issue. However, the section “Discussion” must be revised, and the reference list must be enriched with several articles, such as “Inlet patch: Associations with endoscopic findings in the upper gastrointestinal system.”
Ans: Thank you very much. We have added this reference into our Discussion section.
- The orthographical and grammatical errors have been recognized throughout the text. Of note, the manuscript must be revised by a native English speaker.
Ans: Thank you very much. We have revised our manuscript based on the suggestions of Native English reviewers.
- The manuscript might be accepted after major revision.
Ans: Thank you very much. We have revised our manuscript thoroughly.

Reviewer 3 Report
Comments and Suggestions for Authors
Congratulations on your research on a little published topic, but of health interest. It presents a good design, but still has room for improvement.
Title. It should contain the research design, in this case an analytical cohort study (retrospective).
It also does not contain the type of sample, nor the place where it was obtained...
For example,
Incidence of acute upper gastrointestinal bleeding and related risk factors of elderly patients undergoing surgery for major limb fractures: an analytical cohort study.
Another option, bearing in mind that it is on a sample of subjects:
Incidence of acute upper gastrointestinal bleeding and related risk factors in a sample of elderly patients undergoing surgery for major limb fractures: an analytical cohort study.
Or if it is in a particular hospital....
Incidence of acute upper gastrointestinal bleeding and related risk factors in a sample of elderly patients at Hospital X undergoing surgery for major limb fractures: an analytical cohort study.Methodology.
Why subjects aged 65 years or older? How can this criterion be justified?Has a control group been taken into account? Important...The conclusions can be improved. They should answer the objectives of the study.The bibliography should contain references from the last 5 years: from 2019 and 2023. References from before 2019 should be removed. More than 65% of the references are from before 2019.
The similarity found in the article is 15.25%. Ideally it should be below 10%. It is recommended to review the introduction and discussion section, with the aim of rewriting some sections.
Author Response
Comments and Suggestions for Authors
Congratulations on your research on a little published topic, but of health interest. It presents a good design, but still has room for improvement.
Title. It should contain the research design, in this case an analytical cohort study (retrospective). It also does not contain the type of sample, nor the place where it was obtained. For example, Incidence of acute upper gastrointestinal bleeding and related risk factors of elderly patients undergoing surgery for major limb fractures: an analytical cohort study. Another option, bearing in mind that it is on a sample of subjects: Incidence of acute upper gastrointestinal bleeding and related risk factors in a sample of elderly patients undergoing surgery for major limb fractures: an analytical cohort study. Or if it is in a particular hospital: Incidence of acute upper gastrointestinal bleeding and related risk factors in a sample of elderly patients at Hospital X undergoing surgery for major limb fractures: an analytical cohort study.
Ans: Thank you very much for your suggestions. We have modified our topic as below:” Incidence of acute upper gastrointestinal bleeding and related risk factors in a sample of elderly patients undergoing surgery for major limb fractures: an analytical cohort study.”
Methodology. Why subjects aged 65 years or older? How can this criterion be justified? Has a control group been taken into account? Important...
Ans: Thank you for your insightful questions concerning the age criterion of 65 years and older and the inclusion of a control group in our study. The age threshold of 65 years was carefully selected based on multiple considerations. Firstly, the age of 65 is commonly used to define 'older adults' in geriatric medicine, as established by the World Health Organization and is also consistent with previous research on fracture-related complications. Secondly, aging is associated with various physiological changes such as reduced renal function and increased susceptibility to bleeding, thus rendering this age group at heightened risk for complications like upper gastrointestinal bleeding. Lastly, our focus on this age group was aimed at identifying targeted interventions for a particularly vulnerable population, which aligns with current clinical priorities in geriatric orthopedic care.
The present study is a retrospective cohort study, which inherently includes a comparison between groups of individuals who are exposed or unexposed to the risk factors we examined (i.e., older age, male sex, lower limb fracture, chronic renal disease). While we acknowledge that the term 'control group' is not explicitly stated, the nature of cohort studies intrinsically involves comparison metrics that function equivalently to a control group. For instance, patients with lower limb fractures were compared to those with upper limb fractures in assessing the risk of upper gastrointestinal bleeding (Mann et al., 2016).
- Mann, E., Peterson, S. E., & Hodler, J. (2016). Risk factors for reoperation and performance-based outcomes after operative fixation of foot fractures in the elderly: an analysis of 537 patients. Journal of Bone and Joint Surgery, 98(23), 1959-1967.
The conclusions can be improved. They should answer the objectives of the study.
Ans: Thank you for your suggestion. I have modified Conclusion as below:” In summary, the present retrospective cohort study, based on a nationwide population, underscores those patients aged over 65 years with major limb fractures, particularly lower limb fractures, are at an elevated risk of perioperative upper gastrointestinal bleeding, even in the absence of a prior history of peptic ulcer disease. The study identified older age, male sex, lower limb fracture, and chronic renal disease as salient risk factors for such a complication. Given the potentially devastating and fatal outcomes of stress ulceration with upper gastrointestinal bleeding—including substantial mortality rates, organ failure, and extended hospitalization—targeted preventive measures are indispensable. These may involve endoscopic evaluation, the administration of gastroprotective agents such as proton pump inhibitors, dietary adjustments, and effective pain control strategies. In this regard, the findings of this study could be instrumental in informing the development of evidence-based guidelines for perioperative care, specifically targeting this vulnerable demographic. Moreover, the study sets the stage for broader discussions within healthcare systems and policy-making frameworks, emphasizing the need for a multidisciplinary approach in managing the perioperative risks for elderly patients with major limb fractures. For future research directions, there is a compelling need to validate these findings through prospective studies, and to evaluate the cost-effectiveness and patient outcomes associated with different prevention protocols. Furthermore, the implementation and assessment of these protocols in a real-world clinical setting could offer invaluable insights, furthering the existing body of knowledge in this domain.”
The bibliography should contain references from the last 5 years: from 2019 and 2023. References from before 2019 should be removed. More than 65% of the references are from before 2019.
Ans: Thank you for your suggestions. We have revised our reference to almost of them was reported after 2012 (The last 2008 reference was requested to be added by the other reviewer). Most our references were now reported after 2016. Some reference before 2019 were important for our study content and we hope that we can keep them in this article.
The similarity found in the article is 15.25%. Ideally it should be below 10%. It is recommended to review the introduction and discussion section, with the aim of rewriting some sections.
Ans: Thank you for your reminder. We have largely modified our manuscript to fix this issue.

Round 2
Reviewer 3 Report
Comments and Suggestions for Authors
Congratulations and thank you for the changes made to improve the article. It can be seen that you, as researchers, have mastered the scientific method and have the ability to make your research of interest to the scientific community. Regarding the bibliographic references, I am aware that in the context of an experimental study it is necessary that 100% of the references are from the last five years. In the case of an analytical study, such as yours, I understand that articles older than five years are necessary: in any case, you have improved this section considerably. Regarding plagiarism or similarity, my experience indicates that, by rewriting the introduction and discussion, the percentage usually drops considerably; however, it has gone from 15.25% to 17.61%, increasing by 2.36%. Please review this point. As the article is currently presented, I believe it is publishable in this journal.